# Understanding the Gut–Brain Axis and Its Therapeutic Implications for Neurodegenerative Disorders

**DOI:** 10.3390/nu15214631

**Published:** 2023-10-31

**Authors:** Yadong Zheng, Laura Bonfili, Tao Wei, Anna Maria Eleuteri

**Affiliations:** 1School of Biosciences and Veterinary Medicine, University of Camerino, 62032 Camerino, MC, Italy; yadong.zheng@unicam.it (Y.Z.); laura.bonfili@unicam.it (L.B.); 2School of Food and Biological Engineering, Zhengzhou University of Light Industry, Zhengzhou 450002, China

**Keywords:** gut–brain axis, gut microbiota, gut dysbiosis, neurodegenerative disorders, probiotics

## Abstract

The gut–brain axis (GBA) is a complex bidirectional communication network connecting the gut and brain. It involves neural, immune, and endocrine communication pathways between the gastrointestinal (GI) tract and the central nervous system (CNS). Perturbations of the GBA have been reported in many neurodegenerative disorders (NDDs), such as Alzheimer’s disease (AD), Parkinson’s disease (PD), and amyotrophic lateral sclerosis (ALS), among others, suggesting a possible role in disease pathogenesis. The gut microbiota is a pivotal component of the GBA, and alterations in its composition, known as gut dysbiosis, have been associated with GBA dysfunction and neurodegeneration. The gut microbiota might influence the homeostasis of the CNS by modulating the immune system and, more directly, regulating the production of molecules and metabolites that influence the nervous and endocrine systems, making it a potential therapeutic target. Preclinical trials manipulating microbial composition through dietary intervention, probiotic and prebiotic supplementation, and fecal microbial transplantation (FMT) have provided promising outcomes. However, its clear mechanism is not well understood, and the results are not always consistent. Here, we provide an overview of the major components and communication pathways of the GBA, as well as therapeutic approaches targeting the GBA to ameliorate NDDs.

## 1. Introduction

In recent years, there has been a notable surge in interest surrounding the gut–brain axis (GBA) due to its increasingly recognized role in the regulation of health and the development of diseases, as well as its potential as a therapeutic target [1,2]. A growing body of experimental evidence underscores the profound influence of the microbiota via the GBA on neurodegenerative disorders (NDDs) such as Alzheimer’s disease (AD) and Parkinson’s disease (PD) [3,4]. In this intricate bidirectional communication, the gut microbiota directly or indirectly interacts with the enteric nervous system (ENS) [5], immune system [6], and enteroendocrine system [7], facilitating signal transmission via spinal nerves, the vagus nerve, and the circulatory system to the central nervous system (CNS). This involves a myriad of metabolites and neurotransmitters produced by microorganisms, including peptides, gut hormones, cytokines, and neuroactive substances, suggesting the pivotal role of the gut microbiota in maintaining this communication and various metabolites in homeostasis [8]. In turn, the brain can regulate gastrointestinal motility, microbiota composition, and homeostasis through the nervous and neuroendocrine systems [9]. Moreover, the composition of the gut microbiota influences many aspects of gut and brain function, including the integrity of the intestinal barrier, the stability of the endocrine system, the permeability of the blood–brain barrier (BBB), and the maturation and polarization of microglia and astrocyte cells. Gut dysbiosis may participate in the pathophysiology of various neurological diseases, including NDDs [10]. However, the exact mechanisms through which an altered gut microbiota can impact the CNS remain unclear. This review aims to provide an overview of the current literature on the role of the GBA in NDDs and to better understand this communication pathway, focusing on its major components, communication pathways, and potential therapeutic approaches.

## 2. Gut–Brain Axis

The gut–brain axis (GBA) represents the bidirectional communication between gut microbiota and the brain. This intricate communication involves the coordinated interaction of the nervous system, immune system, and endocrine system (Figure 1). It plays a pivotal role in regulating both physical and mental health while maintaining homeostasis within the GI tract, CNS, and microbial systems. This regulation is achieved through the direct or indirect influence of chemical transmitters, including microbial hormones and metabolites [11,12,13]. Although not fully delineated, the following are some of the main aspects of the GBA and potential therapeutic effects on NDDs.

### 2.1. Human Gut Microbiota

The gut microbiota refers to a dynamic community of microorganisms that inhabit the GI tract of animals, including humans. It primarily consists of bacteria, along with fungi, archaea, parasites, and viruses to a lesser extent [14]. These microorganisms have coevolved with their hosts over millennia, establishing intricate and symbiotic associations [15]. The collective genetic content of all gut microbial genes within an individual, known as the microbiome, represents a genetic repertoire containing a total of 3.3 million genes [15], which is more than an order of magnitude larger than the human genome [16]. Furthermore, the number of bacteria in the human gut is nearly equivalent to the number of human cells, significantly expanding human metabolic capabilities, comparable to the functions of the liver [17].

The human gut microbiota comprises four major phyla (*Firmicutes*, *Bacteroidetes*, *Proteobacteria*, and *Actinobacteria*), along with two minor phyla (*Fusobacteria* and *Verrucomicrobia*), with *Firmicutes* and *Bacteroidetes* representing approximately 90% of the gut microbiota [18,19]. Across the GI tract, variations in microbial density and composition are influenced by chemicals, nutrition, and immunological factors. Specifically, the stomach exhibits an exceedingly acidic pH, while the small intestine also maintains a relatively elevated acidity and features a shorter transit duration, properties that limit microbiota growth and result in few microbiota species [20]. In contrast, the colon/large intestine, characterized by anaerobiosis, a slower passage of food, water for absorption, and undigested food for fermentation, is densely colonized with microbiota [20]. This spatial diversity emphasizes the need to consider anatomical regions in gut microbiota analysis. Within an individual, gut microbiota communities are also dynamically changing entities that can alter their composition and activity in response to intrinsic host factors, such as genetics, age, and general health conditions, as well as extrinsic factors like diet, drugs, lifestyle, physical activity, infection, stress, and geographical location [17]. Variations also exist among individuals: each healthy human possesses a unique gut microbiota. The notion of a “core microbiota” suggests a consistent set of abundant organisms across all individuals [21]. Nevertheless, a greater similarity in the repertoire of microbial genes among individuals, as opposed to the taxonomic profile, indicates that defining the “core microbiota” at a functional level might be more appropriate [21].

The gut microbiota is closely related to human well-being, involving immune response, digestive function, metabolism as well as neurological signaling transmission [22]. Studies suggested that the *Firmicutes*/*Bacteroidetes* ratio serves as an indicator of gut microbiota health. This simple stratification scheme can help in the diagnosis, can determine risk or susceptibility to disease development, and provides a reference for possible therapeutic intervention [23]. Elevated levels of *Firmicutes* and reduced levels of *Bacteroidetes* have been linked to numerous pathological conditions, including type 2 diabetes (T2D), obesity, and dementia [23,24,25].

### 2.2. Autonomic Nervous System

The autonomic nervous system (ANS), encompassing the sympathetic (SNS) and parasympathetic nervous systems (PNS), forms a neural network comprising neurons widely distributed throughout the central and peripheral regions. It regulates involuntary bodily functions such as heartbeat, breathing, and digestion [26,27]. The individual components of the GBA communicate bidirectionally with each other through this network. Considering the GI tract and the CNS, afferent signals from the lumen are transmitted to the CNS or efferent signals from the CNS to the GI tract, through both the SNS and the PNS (including enteric, spinal, and vagal pathways) [28]. In detail, the ANS controls the GI functions, such as intestinal motility and barrier integrity, luminal osmolarity, mucosal secretory, and immune response. These changes in microbiota habitat further affect the relative abundance and diversity of specific microbial taxa. Conversely, gut autonomic nerves can carry sensory information directly to the brain when stimulated by gut microbial metabolites (metabolites interacting with gut ANS synapses), including serotonin, γ-aminobutyric acid (GABA), catecholamines, and precursors of tryptophan [29]. These neurotransmitters can directly interact with the CNS. Visceral information from the gut, transmitted via the ANS, is processed by the CNS, which subsequently triggers an adaptive response with effects on peripheral organs. Thus, the ANS serves as a conduit for immediate and precise neurological responses through its innervation of the target organ.

### 2.3. Vagus Nerve

The vagus nerve is a pivotal component of the PNS, deriving its name from the Latin word meaning “wandering”. It consists of 80% afferent and 20% efferent neurons, which tonically transmit crucial information from visceral organs such as the GI, cardiovascular, and respiratory systems to the CNS (bottom-up signaling) and furnish feedback to the viscera (top-down signaling). Substantial evidence supports the essential roles of vagal nerve pathways in the regulation of appetite, stress responses, inflammation, and cognitive reactions [30].

Vagal afferents establish three distinct categories of connections within the ENS: intraganglionic laminar terminals and intramuscular arrays [31], both ending within the muscular layer, as well as terminal axon endings within the mucosal layer, and a connection with a subset of enteroendocrine cells, referred to as neuropods, which form synapses with vagal neurons. Due to their widespread distribution, types, and expression of a plethora of receptors, vagal afferents are considered multimodal, capable of detecting various molecules such as bacterial byproducts, intestinal hormones, and neurotransmitters. As a result, they are responsive to a range of mechanical, chemical, and hormonal signals [32].

Gut-related signals from vagal afferents travel to the nucleus tractus solitarius (NTS) in the brainstem [33]. This information is then relayed to other nuclei within the brainstem and forebrain structures involved in regulating behavior, emotions, stress, and cognition [34]. Complex multisynaptic pathways originating from the NTS establish connections between visceral information and various regions of the brain. For example, projections to the bed nucleus of the stria terminalis and amygdala play roles in modulating emotions and behaviors, including anxiety, fear, and avoidance behaviors [35]. Similarly, projections to the nucleus accumbens and basolateral amygdala contribute to memory modulation after arousal [36,37]. Additionally, projections to the lateral hypothalamus stimulate feeding behavior, while the NTS also connects to the pituitary and ventral tegmental area, thereby influencing the hypothalamic-pituitary-adrenal (HPA) axis and cognitive functions, respectively. Further projections towards the arcuate nucleus integrate endocrine and behavioral aspects, thereby regulating food intake and satiety [32]. Through direct or multisynaptic projections, the NTS also affects neurotransmitters like norepinephrine and 5-HT. Essentially, the NTS efficiently coordinates the integration of gut–brain feedback via the vagus nerve, serving as a hub for GBA signaling [32].

The vagus nerve plays a pivotal role in promoting neurogenic and neurotrophic signaling pathways. In fact, animal studies have demonstrated that the disruption of the vagus nerve reduced neurogenesis [38] and activated microglia in the hippocampus, resulting in aberrations in stress response and cognition, and causing anxiety- and fear-related behaviors [39]. Conversely, vagus nerve stimulation enhanced hippocampal neurogenesis, regulated the release of neurotransmitters [40], and increased hippocampal brain-derived neurotrophic factor (BDNF) expression, thereby improving synaptic plasticity, learning, and memory [41].

### 2.4. Enteric Nervous System

The enteric nervous system (ENS), a significant component of the ANS, is situated within the GI tract and constitutes a complex mesh of 200 to 600 million neurons facilitating the control of gut functions, including motor activity, secretion, absorption, and immune defense, playing a pivotal role in maintaining gut homeostasis and interacting with the microbiota and host systems [42]. The ENS is anatomically composed of two ganglionated plexuses, the myenteric and submucosal plexus, containing nitrergic and cholinergic neurons [43]. Intrinsic neurons of the ENS commonly communicate with the CNS through the PNS, mainly the vagus nerve, and the SNS, such as the prevertebral ganglia. These complicated intrinsic and afferent neural signals create avenues for factors originating from the gut lumen, potentially encompassing the microbiota, to influence not only intestinal functions but also the CNS. The structure and neurochemistry of the ENS resemble that of the CNS, which is why it often is referred to as the “second brain”, and thereby, any mechanisms implicated in CNS dysfunction may also result in ENS dysfunction or vice versa [44,45].

The gut microbiota significantly influences ENS development and function through the activation of pattern recognition receptors (PRRs) such as toll-like receptors TLR-2 and TLR-4, which recognize microbial lipopolysaccharide (LPS), peptidoglycan, and viral RNA. Studies in TLR-deficient mice have demonstrated changes in ENS functions, including reduced stool output, water content, and gut motility [46,47]. Germ-free (GF) mice have exhibited a disrupted ENS structure, reduced enteric neurons, compromised gut motility, and impaired sensory signaling [42]. GF mice also exhibit abnormal neurochemistry and insufficient influx of enteric glial cells into the intestinal mucosa [48]. These observations are mirrored in mice with antibiotic-induced gut microbial dysbiosis [47]. The gut microbiota promotes serotonin biosynthesis by enterochromaffin cells, vital for mucosal and platelet function [49]. Additionally, the gut microbiota can also produce neurotransmitters and metabolites like GABA, histamine, catecholamines, acetylcholine, and short-chain fatty acids (SCFAs), further shaping ENS activity. On the contrary, the ENS seems to possess the ability to exert an influence on the gut microbiota. An investigation utilizing a transgenic zebrafish model with impaired ENS function unveiled a notable alteration in the composition of the GI microbiota, shifting it towards a proinflammatory microbial profile. Intriguingly, the introduction of ENS precursors through transplantation reversed the microbiota back to its normal state [50]. These findings suggest a bidirectional interaction between the gut microbiota and the ENS, highlighting the complex interplay between these systems. Moreover, ENS has now been implicated in NDDs [51], including AD and PD, typically considered primary CNS conditions. This underscores once again the vital role of ENS in intricate communication between the gut and the brain.

### 2.5. Neuroendocrine Hypothalamic–Pituitary–Adrenal Axis

The hypothalamic–pituitary–adrenal (HPA) axis is considered an essential neuroendocrine pathway integral to GBA communication, orchestrating physiological adaptation to stress. During stress, the hypothalamus initiates the synthesis and secretion of corticotropin-releasing hormone (CRH), which serves as the primary regulatory factor for the HPA axis. CRH subsequently traverses to the anterior pituitary gland, where it binds to its corticotropin receptor, thereby inducing the release of adrenocorticotropic hormone (ACTH) into the systemic circulation. ACTH stimulates the adrenal gland to synthesize and secrete glucocorticoids (cortisol in humans and corticosterone in rodents), serving as downstream effectors of the HPA axis. These glucocorticoids regulate physiological alterations through ubiquitously disseminated intracellular receptors to meet metabolic, physical, and psychological demands under stress. Nevertheless, both excessive and insufficient activation of the HPA axis leads to psychophysiological disorders [52,53].

Specifically, appropriate levels of glucocorticoids are essential for proper neurodevelopment, and cognitive processes such as learning and memory [54]. Experimental models investigating stress have demonstrated a correlation between the HPA axis and alterations in microbiota composition as well as its metabolites [55]. Conversely, the microbial modulation of the HPA axis also affects glucocorticoid concentrations [30]. Modulating gut microbiota through the administration of probiotics and prebiotics has been shown to ameliorate the stress-dependent increase in corticosterone levels [56,57]. Considering the widespread distribution of glucocorticoid receptors across multiple organs, including the GI tract and the CNS, as well as on various cells such as neurons, epithelial cells, immune cells, and endocrine cells, glucocorticoids can affect both gut and brain functions through multiple pathways including neural, metabolic, immunological, and endocrine pathways. Prolonged stress can induce the HPA axis dysregulation. In fact, it has been observed that increased cortisol is associated with cognitive decline and increased AD risk, with elevated cortisol in dementia patients involving interactions between inflammation, neurotransmitters, and oxidative stress [54].

### 2.6. Neurotransmitters

Neurotransmitters mediate intercellular signal transduction across diverse neuronal types and glial cells, influencing learning, memory, emotion, and movement. Neurotransmitters can be categorized into excitatory neurotransmitters, such as glutamate, acetylcholine, norepinephrine, and dopamine, as well as inhibitory neurotransmitters, such as GABA, glycine, and serotonin [58]. Their synthesis and regulation are orchestrated by neurons and glial cells through specific enzymes, and their disruption is implicated in NDDs and psychiatric disorders, including AD, PD, depression, and anxiety [59].

Interestingly, emerging evidence underscores that certain members of the microbiota are capable of producing enzymes or metabolites that facilitate the synthesis of neurotransmitters or their precursors, thereby influencing brain function [60,61]. This introduces an additional communication pathway within the GBA. However, only a limited number of neurotransmitters can cross the BBB directly and act on the CNS. They can indirectly modulate brain activity via local interaction with the ENS or rapid signaling to the brain via the vagus nerve [62]. Moreover, some neurotransmitter precursors can traverse the BBB through the carrier system present in capillary endothelial cells [63]. Subsequently, neurotransmitter-producing cells uptake these precursors, allowing for their conversion into functional neurotransmitters through a series of intermediary steps facilitated by host enzymes. Specific bacterial strains have been identified as responders to or producers of catecholamines. For example, the genus *Escherichia* is known to produce norepinephrine [64], and *Bacillus* can biosynthesize norepinephrine and dopamine, both catecholamines that regulate emotion, cognition, and gut motility. *Staphylococcus* inhabited in the human gut also has been reported to express staphylococcal aromatic amino acid decarboxylase, able to convert precursor L-3,4-dihydroxy-phenylalanine (L-DOPA) into dopamine [60]. *Lactobacillus* spp. [65] and *Escherichia* spp. [66] have been reported to synthesize GABA, which regulates and coordinates neuronal signaling in the hippocampus, thereby influencing cognitive processes. Serotonin is involved in regulating cognition, GI secretion, and motility, as well as circadian rhythm, and is almost 90% synthesized by enterochromaffin cells [67]. Interestingly, spore-forming bacteria, particularly *Clostridia*, have been linked to the promotion of serotonin biosynthesis by enhancing the gene expression of tryptophan hydroxylase 1, the rate-limiting enzyme for serotonin synthesis [49], and certain metabolites, such as SCFAs and bile acids, can influence this synthesis process. Furthermore, *staphylococci* can use amino acid decarboxylase to decarboxylate the precursor 5-hydroxytryptophan into serotonin [60]. *E. coli* and *Morganella morganii* are some of the bacteria known to produce biogenic amines like histamine [68], which is responsible for regulating wakefulness as well as various immune functions, thereby potentially affecting the host immune system. Acetylcholine, a crucial cholinergic neurotransmitter, functions as a local modulator within both the central and peripheral nervous systems, facilitating excitatory signals between neurons [69]. Its dysregulation is closely implicated in AD pathology [70] and can be synthesized by a variety of bacteria including *Bacillus subtilis*, *Lactobacillus plantarum*, *E. coli*, and *Staphylococcus aureus* [71,72]. Research conducted on germ-free animals and the utilization of antibiotics to deplete the gut microbiota have revealed significant alterations in neurotransmitters and their receptor levels in the brain, GI, and blood [73,74]. These collective findings underscore the intricate interplay between the gut microbiome and host neurotransmitter levels. It is plausible that the gut environment may exert direct and indirect effects on neuronal activity and cognitive function within the brain through such interconnected pathways.

### 2.7. Immune System Pathway

The immune system plays a crucial role in distinguishing between “harmful” and “harmless” signals and orchestrating appropriate responses, especially in the GI tract, where the immune system constantly interacts with microorganisms through innate and adaptive immunity [75]. Enterocytes express innate immune receptors and can release cytokines and chemokines, while gut-associated lymphoid tissue (GALT) utilizes lymphocytes to generate a more specific immune response involving immunoglobulins.

Notable immune receptors include PRRs such as TLRs, which specifically identify microorganism-associated molecular patterns (MAMPs). Examples of these patterns include LPS and polysaccharide A for Gram-negative bacteria and peptidoglycan for Gram-positive bacteria [5,76]. This recognition mechanism enables immune system cells to detect and respond to microbial presence, discern alterations in bacterial balance, and uphold gut homeostasis. Specifically, the activation of immune cells initiates a cascade, recruiting inflammatory mediators [77], cytokines, and chemokines that act as chemical messengers to facilitate communication between the immune system, gut microbiota, and CNS. Maintaining a balance between anti-inflammatory and proinflammatory cytokines and chemical messenger is a key determinant of an appropriate host defense against infection or tissue damage. Proper intestinal immune cytokine production maintains intestinal homeostasis, which in turn affects local microbial concentrations, while excessive cytokines may cross the BBB from the systemic circulation and directly affect brain function. Dysregulated microbiota can compromise the integrity of both the intestinal barrier and the BBB, allowing the infiltration of microbes and their products into the CNS, triggering the activation of the brain’s immune system such as resident immune cells like microglia, leading to a proinflammatory state [78]. It is noteworthy that chronic inflammation is associated with cognitive impairment and behavioral alterations.

Gut microbiota depletion induced by antibiotic treatment or GF mice exhibit systemic and CNS immune system responses [79,80]. The microbiota can modulate neuroinflammation by affecting monocyte migration from the periphery to the CNS. A further analysis found that this monocyte migration was mediated by tumor necrosis factor-alpha (TNF-α), a cytokine produced by microglia [81], which can be reversed by the administration of probiotics [82]. A study on recombination activation gene 1 (Rag1) transgenic mice lacking lymphocytes revealed cognitive and anxiety-related behavioral changes, highlighting the crucial role of the immune system in the GBA, and this impairment of brain function was reversed by a probiotic combination—*L. rhamnosus* and *L. helveticus* [83]. Thus, probiotic treatment holds promise for alleviating adaptive immune damage and resulting behavioral changes.

### 2.8. Enteroendocrine Signaling

Enteroendocrine cells (EECs) are distributed widely throughout the GI tract, comprising only 1% of the epithelial cell population within this milieu. Nonetheless, they collectively form the body’s largest endocrine organ and wield pivotal regulatory effects on bidirectional communication between the gut and the brain [84,85]. Currently, ten distinct EEC subtypes have been identified [86], traditionally characterized based on their secretion of gut hormones. These include K-cells (glucose-dependent insulinotropic polypeptide (GIP)), I-cells (cholecystokinin (CCK)), and L-cells (glucagon-like peptide-1 (GLP-1)) [87]. However, recent evidence suggests that EECs are more complicated multihormonal cells, largely influenced by their location and maturation within the gut [85].

EECs primarily exhibit an open-type cell morphology, characterized by a bottleneck shape with an apical membrane in direct interaction with the lumen and a basolateral membrane in proximity to blood vessels and innervating neurons [84]. EECs exhibit a high expression of chemosensory components, including nutrient transporters and nutrient-specific G protein-coupled receptors (GPCRs). This unique configuration enables the EECs to sense changes in the gut lumen, including nutrients, as well as gut microbiota and their metabolites. They act as initial messengers to help the host maintain metabolic processes such as energy and glucose homeostasis, as well as behavioral responses, such as food intake [84,88]. Specifically, in response to mechanical, chemical, or neural stimulation, the influx of intracellular calcium triggers the release of gut peptides via the basolateral membrane into the extracellular milieu, where these gut peptides activate vagal afferent neurons that send signals to the NTS and then cascade to higher-order cerebral domains (refer to the vagal nerve system section for details). Vagal afferent neurons can also be activated through the ENS via gut-derived neurotransmitters such as 5-HT. Additionally, gut peptides can enter the blood circulatory system and signal directly to the NTS [84].

While more than 20 gut peptides have been identified within the GI tract, most of these play a central role in modulating energy and glucose homeostasis [89]. Several gut hormones, including ghrelin, peptide YY (PYY), GLP-1, GIP, and CCK, are implicated in various aspects of gut–brain crosstalk; all of these hormones are released in response to food intake and mutually regulate with the brain [89,90]. For instance, GLP-1 has been demonstrated to enhance hippocampal synaptic plasticity, improve learning, memory, and motor functions, as well as mitigate neuroglial cell activation, suggesting potential neuroprotective effects in NDDs such as AD and PD [91]. Ghrelin has been described as an important link connecting metabolism, aging, and NDDs, affecting glucose and lipid metabolism, and influencing memory and learning consolidation [92].

### 2.9. Blood–Brain Barrier

The BBB is a dynamic interface that separates the CNS from the systemic circulation, maintaining brain homeostasis. Functioning as a selective barrier, it allows essential nutrients, oxygen, and waste products to pass while preventing the entry of toxins, pathogens, and harmful molecules. The disruption of the BBB integrity is implicated in the pathology and progression of NDDs. The BBB is structurally organized by endothelial cells lining cerebral microvessels, along with tight junction proteins, pericytes, basement membranes, and glial cells.

Endothelial cells in the CNS possess unique properties with low rates of transcytosis, higher numbers of mitochondria, a reduced expression of leukocyte adhesion molecules, and an elevated expression of various transporters and enzymes [93]. These features enable the selective movement of solutes into and out of the CNS parenchyma, maintaining a stable microenvironment for optimal neuronal function [94]. The integrity of the BBB is further maintained by tight junction proteins, which consist of transmembrane proteins such as occludin, claudins, and junctional adhesion molecules (JAMs), also involving the recruitment of various membrane-associated cytoplasmic proteins such as zonula occludens (ZO) and actin cytoskeleton [95]. Accumulating evidence indicates a correlation between alterations in tight junction proteins, BBB dysfunction, and the progression of NDDs [96]. Pericytes, located on the abluminal surface of the endothelial cell wall, are involved in various physiological processes associated with BBB maintenance, such as angiogenesis, immune cell infiltration, the modulation of the extracellular matrix, wound healing, and the regulation of blood flow. Pericytes also regulate the BBB permeability through interactions with endothelial cells and the secretion of factors that support BBB function. Astrocytes, with their endfeet ensheathing blood vessels and serving as a cellular link between the CNS and blood vessels, provide metabolic support to endothelial cells. They also secrete factors that modulate tight junction integrity and transport activity, thus contributing to the maintenance of the BBB integrity and functionality [93].

Postmortem and MRI imaging analysis have confirmed the impairment of the BBB in AD patients [97]. Interestingly, gut-derived signals, including microbial metabolites such as LPS, SCFAs, trimethylamines (TMAs), and vitamins, have been shown to modulate the BBB permeability. The dysregulation of the gut microbiota observed in NDDs may contribute to the BBB dysfunction and disease progression [98]. Systemic immune activation has been identified as a potential cause of the BBB integrity damage, as demonstrated by experiments involving the injection of LPS into animals, resulting in a significant 60% increase in BBB permeability [98]. A targeted manipulation of the gut microbiota has been found to increase the expression of claudin-5 and occludin, resulting in a decreased BBB permeability. Conversely, GF mice exhibit a more permeable BBB compared with mice with a typical composition of gut microbiota, and implanting a normal microbiota into GF mice partially restored the barrier function [99,100].

### 2.10. Intestinal Barrier

The intestinal barrier serves as a semipermeable surface responsible for the intricate selective functions of the gut. It facilitates the absorption of essential nutrients and immune surveillance while concurrently restricting the transport of pathogenic molecules and microorganisms. Both structural and molecular components of the barrier interplay in a dynamic manner to fulfill the complex task and maintain intestinal integrity and immune homeostasis [101].

Structurally, the intestinal barrier is primarily composed of the outer mucus layer, a continuous monolayer formed by the epithelial cells, and the inner lamina propria carrying the adaptive and innate immune systems [102]. The outer mucus layer forms a gel-like sieve structure of highly glycosylated mucins, such as mucin 2, which covers the epithelium and serves as the first line of physical defense against a direct contact of external molecules and bacteria with epithelial cells [103]. Immune regulators such as antimicrobial peptides (AMP) and secretory immunoglobulin A (IgA) molecules are also released and distributed in the mucus gel as immune sensing and regulatory proteins [104]. The outer mucus layer provides a habitat for the gut commensal microbiota and offers nutrients (glycans). The composition of the mucus layer can influence the microbiota, while the microbiota can shape the mucus gel. Beneath the mucus layer, the continuous and polarized monolayer of intestinal epithelial cells is constructed by five distinct cell types: goblet cells, absorptive enterocytes, enteroendocrine cells, microfold cells, and Paneth cells, all originating from a reservoir of pluripotent stem cells in the crypts [101]. They are tightly connected to each other through junctional complexes. Specifically, tight junctions are located at the apical side of the cells and serve to seal the intercellular space, regulating the transportation of small molecules and ions. They are composed of transmembrane proteins, such as claudins and occludins, as well as peripheral membrane proteins, such as zonula occludes (ZO)-1 and ZO-2, along with regulatory proteins. Adherent junctions (AJ), situated beneath the tight junctions, together with desmosomes, form strong cell-adhesion bonds to maintain the integrity of the intestinal barrier [105]. The inner lamina propria lies behind the epithelium and consists of cells of the adaptive and innate immune systems, including T cells, B cells, dendritic cells, and macrophages, which are involved in the immune defense mechanisms of the intestinal barrier.

Any damage to the structure and composition of the intestinal barrier may dramatically affect its functionality. A dysfunction of the intestinal barrier has been linked to a range of human diseases, including GI tract diseases such as irritable bowel syndrome [106], as well as extraintestinal disorders including type 1 diabetes [107], obesity [108], AD, PD, and depression [109]. It is widely postulated that the breakdowns in the intestinal barrier and the unregulated movement of antigens and pathogenic molecules across the intestinal epithelium may pose a challenge to the immune system of vulnerable individuals, disrupting the equilibrium between the host and microbial communities, such as potentially triggering inflammatory responses in the GI tract or even in distant organ systems [110].

## 3. Gut Dysbiosis and Neurodegenerative Disorders

### 3.1. Gut Dysbiosis

Gut dysbiosis refers to an imbalance in the gut microbiota, characterized by a decrease in microbial richness, abundance, and the loss of beneficial bacteria such as *Bacteroides* and *Firmicutes*, and an increase in pathogenic bacteria such as *Prevotellaceae* and *Enterobacteriaceae*. This imbalance can have negative effects on the host’s health, including metabolic disorders, endogenous intoxication, systemic inflammation, and reduced essential metabolites [111].

Dysbiosis can stem from a multitude of factors, encompassing an imbalanced dietary regimen predominantly composed of refined constituents, inadequate dietary fiber incorporation, elevated alcohol consumption, exposure to exogenous substances and contaminants, the administration of pharmaceutical agents including antibiotics, persistent psychological stress, sleep deprivation, bacterial or viral infections, as well as various medical conditions such as metabolic disorders notably T2D, and NDDs [111].

Several potential biomarkers of dysbiosis have been employed in disease diagnosis. When combined with clinical and other biomarkers, microbiome-based biomarkers can improve the precision of disease classification [112]. Notably, the detection of urolithin (a class of metabolites derived from colonic microbial degradation of dietary fiber in the human gut) in urine provides a noninvasive method for identifying gut dysbiosis and inflammation in PD, while reduced levels of *Roseburia species* show promise as a PD marker [113]. Microbial metabolites like indoxyl sulfate could serve as diagnostic biomarkers for dysbiosis and neurological conditions and an abundance of *Enterobacteriaceae* may predict poststroke cognitive impairment [112].

### 3.2. The Role of Gut Dysbiosis in the Pathophysiology of Neurodegenerative Disorders

Recent studies have shown that gut microbiota dysbiosis can lead to an increased permeability of both the intestinal barrier and the BBB [114], along with alterations in intestinal mucus and the translocation of gut microbes and their metabolites. These changes contribute to the induction of a state of toxic inflammation [115], causing a series of disturbances in physiological homeostasis, including oxidative stress, pathological protein aggregation, abnormal proteolysis, neuroinflammation, neuronal death, and altered brain morphology. All of these gut dysbiosis-mediated dysfunction in the GBA signaling is linked to the progression of the neurodegeneration, ultimately resulting in behavioral abnormalities and cognitive impairment.

The neurodegeneration process caused by gut dysbiosis is highly complex, with various physiological changes being interrelated (see Figure 2). Specifically, the primary pathological mechanism triggered by gut dysbiosis increases the permeability of the intestinal barrier, and the gut microbiota and their metabolites constantly interact with PRRs expressed in various host cells, including intestinal epithelial cells, immune cells in peripheral blood, as well as neurons and glial cells within the CNS. These PRRs, such as TLRs, formyl peptide receptors, and the receptor for advanced glycation end products, can recognize pathogen-associated molecular patterns (PAMPs) or danger-associated molecular patterns (DAMPs), which are highly conserved microbial structures including nucleotides, proteins, and LPS [116]. Consequently, active immune signaling pathways, such as the inflammasome, nuclear factor kappa B (NF-κB), myeloid differentiation primary response 88 (MyD88)-dependent pathways, and type 1 interferon, lead to chronic inflammation [116]. Chronic neuroinflammation promotes the aggregation of misfolded proteins around neurons, disrupting neuronal function, permeability, and synaptic integrity [117]. These changes result in neuronal death and the release of misfolded neurotoxic aggregates, further exacerbating neuroinflammation.

Furthermore, the decreased presence of beneficial gut microbiota due to dysbiosis results in impaired microbial metabolism of neuroactive substances (such as tryptophan, SCFAs, and serotonin levels) [118,119]. Conversely, an increase in harmful microbiota elevates the production of toxic metabolites. In patients with AD, PD, and ALS, there is an increased abundance of cyanobacteria in the gut, which secretes higher levels of β-N-methylamino-L-alanine (BMAA), an excitotoxin that binds to metabotropic glutamate receptor 5 and depleting the major antioxidant glutathione [120]. This results in the excessive production of reactive oxygen species (ROS) and reactive nitrogen species (RNS) in the brain, which in turn activates microglia and astrocytes, suggesting a direct link between gut dysbiosis and oxidative stress in NDDs [120].

Emerging research highlights a bidirectional link between gut microbiota and autophagy [121]. Autophagy plays a role in degrading invading pathogens, such as *Salmonella enterica* and *E. coli*, and is involved in antigen presentation and lymphocyte development. The disruption of autophagy in the gut can exacerbate gut dysbiosis. In turn, gut dysbiosis can lead to chronic inflammation and oxidative stress, impairing autophagic clearance processes in both the gut and the brain. Gut dysbiosis also contributes to fibril formation. In PD, α-synuclein inclusion bodies are observed in the brainstem and GI tract. Transplanting fecal microbiota from PD patients to α-synuclein overexpression mice worsened inclusion bodies and PD symptoms compared to mice receiving healthy donor microbiota [122]. Since the microbiota was found to be necessary for exacerbating α-synuclein pathology, gut dysbiosis may trigger and promote α-synuclein fibril formation, dissemination, and disease pathology. Furthermore, the curli protein, one of the major components of the bacterial extracellular matrix, was shown to accelerate fibrilization by cross-seeding and aggregation of α-synuclein and β-amyloid [123].

Interestingly, numerous studies have revealed changes in the microbiome associated with specific NDDs. For example, one study reported a decrease in *Prevotellaceae species* and an increase in *Enterobacteriaceae* abundance in the feces of PD patients [124]. *Prevotellaceae* play a role in producing mucin and SCFAs through dietary fiber fermentation. This reduction could increase gut permeability, allowing endotoxins to enter the bloodstream, and potentially encouraging α-synuclein misfolding and expression. The prevalence of *Enterobacteriaceae* has been positively correlated with the severity of postural instability and gait issues. An increase in *Enterobacteriaceae* raises serum LPS, triggers systemic inflammation, disrupts the BBB, and contributes to α-synuclein deposition and dopamine neuron loss in the substantia nigra [125]. Another study found higher levels of proinflammatory cytokine-producing bacteria like *Ralstonia*, *Proteobacteria*, and *Enterococcaceae*, along with reduced levels of anti-inflammatory butyrate-producing bacteria including *Faecalibacterium*, *Coprococcus*, *Roseburia*, and *Blautia* in PD patients’ mucosa and stool samples [126]. Similar patterns have also been observed in AD, where studies have reported increased levels of proinflammatory bacteria like *Escherichia/Shigella* spp. and decreased levels of anti-inflammatory bacteria such as *Eubacterium rectale* spp. in the fecal microbiome [127]. Collectively, gut dysbiosis can disrupt the intestinal and blood–brain barriers, alter the spectrum of microbial metabolites, induce intestinal, systemic, and neural inflammatory responses, trigger autophagic defects, and lead to increased oxidative stress and pathological protein aggregation. Gut dysbiosis disturbs these communication pathways within the GBA, involving neural, immune, and metabolic functional impairments, ultimately contributing to the pathophysiology of NDDs.

## 4. Therapeutic Approaches Targeting Gut–Brain Axis

The link between gut dysbiosis and NDDs suggests that dietary interventions targeting gut dysbiosis could be a promising strategy for treating symptoms and slowing down the neuroinflammatory and degenerative processes in NDDs. To establish eubiosis and promote overall gut health, complementary nutritional interventions aimed at modulating gut microbiota composition and associated metabolites could complement existing therapeutic approaches. These interventions include various dietary plans, prebiotics, probiotics, synbiotics, and fecal microbiota transplantation (FMT) (Figure 3), which have demonstrated positive effects in reversing gut dysbiosis and promoting a healthy gut state [128].

### 4.1. Diet

Diet can affect gut microbiota composition, which plays a critical role in maintaining host homeostasis. The Mediterranean diet (MD) emphasizes fruits, vegetables, legumes, and cereals and is considered a healthy dietary receipt. One trial found that adherence to the MD slowed the progression of AD by 1.5 to 3.5 years, and beneficial effects of the MD may be mediated through changes in gut microbiota and its anti-inflammatory characteristics [129].

A high consumption of plant-based foods consistent with the MD diets modulates the gut flora, increasing SCFAs in feces and decreasing trimethylamine N-oxide (TMAO) in urine [130]. Supplementation with microbiota-accessible carbohydrate (dietary fiber) improved gut dysbiosis, intestinal barrier integrity, and systemic inflammation in mice, further reducing neuroglial activation and synaptic dysfunction, but broad-spectrum antibiotic depletion reversed these effects [131]. Moreover, a healthy diet can avoid obesity and improve insulin resistance, which are considered potential risk factors for NDDs, such as AD.

Polyphenols, rich in fruits and vegetables, exhibit diverse physiological roles, can regulate ROS to prevent oxidative stress, modulate the autophagy pathway to reduce apoptosis, affect gut microbiota composition to improve the integrity of the intestinal barrier, and repair gut inflammation [132]. In animal models of AD, resveratrol, a polyphenol found in grapes and wine, improved cognition and reduced neuroinflammation. The potential to decrease pathological protein aggregation has also been observed, including Aβ plaques and neurofibrillary tangles (NFT) [133]. In a rat model of PD, resveratrol prevented dopaminergic neuron loss and reduced oxidative stress, lipid peroxidation, and protein carbonyl [134].

High-sugar, high-fat diets, and alcohol consumption all have detrimental effects on NDDs. However, various dietary components, such as polyunsaturated fatty acids, antioxidants like blueberry polyphenols, curcumin, sulforaphane, resveratrol, and salvionic acid, as well as caloric restriction, are beneficial for health [135]. This suggests that dietary preferences in different populations may be associated with the risk of developing NDDs. A 2014 meta-analysis, based on five prospective studies, revealed that individuals adhering to the MD had a 33% lower risk of AD and mild cognitive impairment (MCI) compared to those with a lower adherence [136]. Further research within the Hellenic Longitudinal Investigation of Aging and Diet cohort, comprising 1865 participants, 90 of whom had dementia and 223 with MCI, confirmed the protective effect of an MD adherence against dementia and cognitive decline [137].

### 4.2. Prebiotics

Prebiotics are defined as indigestible dietary components in the upper part of the human GI tract but are selectively utilized by beneficial gut microbes in the large bowel, thereby promoting host health [138]. According to the structure of prebiotics, they can be divided into fructans including inulin and fructo-oligosaccharide or oligofructose (FOS), galacto-oligosaccharides (GOS), resistant starch (RS) known to resist digestion in the upper GI tract, glucose-derived oligosaccharides, and noncarbohydrate oligosaccharides. Prebiotics provide an energy source to specific gut bacteria, altering their composition and activity. They can be fermented through a complex series of processes, producing byproducts that encourage cross-feeding between microorganisms. The acidic fermentation products produced by metabolizing prebiotics that reduce intestinal pH, affecting gut microbial composition and abundance, mainly providing a favorable environment for beneficial bacteria, such as *Lactobacilli* and *Bifidobacteria*, and inhibiting pathogenic bacteria [139]. The primary by-products produced from prebiotics are SCFAs. These small molecules can permeate gut enterocytes and enter the bloodstream, enabling prebiotics to have an impact not only on the GI tract but also on distant organs and systems, including the brain [140].

RS has been shown to maintain bacterial abundance and improve lipid metabolism and intestinal function [141]. Recently in a murine model of aging, RS from pinto beans, black-eyed peas, lentils, and chickpeas administration improved SCFAs production while reducing bile acids and cholesterol, modulated the gut metabolomic pool, therefore mitigating obesity-related metabolism disorders [142]. FOS as a prebiotic ingredient usually found in fruits and vegetables, promotes the growth of healthy gut microbiota and maintains microbial diversity and stability. FOS can also mitigate neuronal apoptosis and brain tissue swelling and improve neurotransmitter synthesis and release. It has also been demonstrated to ameliorate cognitive damage and neurodegeneration in AD mice by regulating the gut microbiota and activating the GLP-1 pathway [143]. In another study, the administration of FOS derived from Morinda officinalis demonstrated significant improvements in behavioral parameters, particularly in terms of learning and memory abilities, in rats with AD induced by D-galactose and Aβ1-42. These improvements were accompanied by a reduction in oxidative stress and inflammation levels and an increase in the levels of crucial neurotransmitters, including 5-HT, 5-hydroxy indole acetic acid, acetylcholine, and dopamine. These positive effects were attributed to the ability of FOS to modulate both gut and brain homeostasis [144].

### 4.3. Probiotics

Probiotics refer to live nonpathogenic microorganisms that confer health benefits when consumed in adequate amounts. Some of the most commonly used probiotics include *Bifidobacterium*, *Lactobacillus*, *Bacillus*, *Streptococcus*, and *Enterococcus* [145,146]. Studies have demonstrated that probiotics confer their benefits through various mechanisms: they regulate gut microbial populations and reduce pathogens colonization and invasion; they increase epithelial cell proliferation and differentiation to reinforce intestinal barrier and reduce immunomodulation; probiotics also produce beneficial chemicals such as SCFAs with anti-inflammatory and neuroprotective effects that enter the circulatory system and cross the BBB, and modulate CNS immune cell activity, inflammatory cytokines, BBB integrity, and neurogenesis, thereby prompting brain health [146,147]; they stimulate the synthesis and release of neurotransmitters, affecting BDNF levels, synaptic plasticity, and neuronal function in NDDs [148].

A probiotic formulation of *Lactic acid bacteria* and *Bifidobacteria*, named SLAB51, strategically modulated gut microbiota composition, increasing *Bifidobacterium* spp., and decreasing *Campylobacterales*, with an increased production of SCFAs and neuroprotective gut peptide hormones, which contribute to reducing Aβ aggregates and preventing cognitive decline. Chronic supplementation with SLAB51 restored the functionality of neuronal proteolytic pathways, reduced cerebral oxidative stress via SIRT1-dependent pathways, and improved glucose homeostasis in the 3xTg-AD murine model of AD [149,150]. Akbari et al. examined the cognitive effects of supplementing AD patients with *Lactobacillus acidophilus*, *Lactobacillus casei*, *Bifidobacterium bifidum*, and *Lactobacillus fermentum*. Supplementation with these probiotics improved patients’ cognitive performance and metabolic parameters, such as insulin sensitivity and serum triglyceride levels [151]. Long-term administration of the probiotic *Lactobacillus paracasei* K71 prevented age-dependent cognitive impairment, with the proposed mechanism involving the upregulation of BDNF levels in the hippocampus [152]. Probiotic-4, containing *Lactobacillus*, *Bifidobacterium*, *Typhimurium*, and *Eosinophilus*, was found to ameliorate age-related BBB and intestinal barrier disturbances. It also reduced plasma and brain LPS and proinflammatory markers such as IL-6 and TNF-α and inhibited TLR4 and NF-κB inflammatory pathways in an ALS mice model [153].

### 4.4. Synbiotics

Synbiotics are defined as specialized formulations of prebiotics and probiotics in which the prebiotics selectively promote the growth and metabolic activity of probiotics, enhance their viability and benefits, positively affect the host’s microbiota composition, and increase the abundance of beneficial microbes in the GI tract, thereby potentially conferring health benefits on the host [154]. The composition used in synbiotics should be thoughtfully selected to support the viability of probiotics in the GI tract. Research findings have suggested that the use of synbiotics is more effective compared to the separate administration of probiotics or prebiotics alone [155].

A novel symbiotic comprising prebiotic polymannuronic acid and probiotic *Lacticaseibacillus rhamnosus* GG promoted their respective neuroprotection against PD. Specifically, the administration of polymannuronic acid or *Lacticaseibacillus rhamnosus* GG alone or in combination for 5 weeks prevented the loss of dopaminergic neurons, improved movement, activity, muscle strength, and increased the expression of the tyrosine hydroxylase gene/protein in the midbrain and striatum of PD mice. Remarkably, the synbiotic showed significantly superior neuroprotective effects [156]. Another innovative form of synbiotic, comprising three metabolically active probiotics—*Lactobacillus plantarum* NCIMB 8826, *Lactobacillus fermentum* NCIMB 5221, and *Bifidobacteria longum spp. infantis* NCIMB 702255—along with a polyphenol-rich prebiotic, acted through the GBA to enhance survivability and improve motility, reduce Aβ deposition, and decrease acetylcholinesterase activity, delaying the onset of AD in a Drosophila model [157]. A randomized controlled trial was conducted to examine the effects of a 12-week synbiotic supplementation on antioxidant capacity, quality of life, as well as mental and fatigue status in patients with PD. The findings revealed that synbiotic supplementation effectively augmented the antioxidant capacity, alleviated depression symptoms, and ameliorated cognitive impairment and activities of daily living in the patients [154].

### 4.5. Fecal Microbiota Transplantation

Fecal microbiota transplantation (FMT) aims to restore a healthy gut microbiome and enhance gut microbiota diversity and function by transferring prescreened donor feces into the patients’ GI tract. Many studies have focused on the potential of FMT in the treatment of inflammatory bowel disease, metabolic disorders, and NDDs [158,159,160].

In APP/PS1 transgenic mice, Harach et al. found that 16S rRNA sequencing of conventionally raised APP/PS1 mice showed significant gut microbiota alterations compared to wild-type mice. In contrast, GF APP transgenic mice had significantly lower levels of Aβ than control mice. FMT from conventionally raised APP/PS1 exacerbated cerebral Aβ pathology in GF APP/PS1 mice, while FMT from wild-type mice did not [161].

The frequent transfer and transplantation of fecal microbiota from wild-type mice to ADLP${}^{APT}$ mice have been found to reduce amyloid plaque and NFT formation, glial responses, and cognitive impairment. FMT has also been shown to reverse the abnormal expression of intestinal macrophage activity-related genes and the increase in circulating blood inflammatory monocytes in ADLP${}^{APT}$ recipient mice [162]. Additionally, in a study involving PD patients, researchers used FMT to treat 11 PD patients, reconstructed the gut microbiota, and improved both motor and nonmotor symptoms in the patients [163].

The availability of well-organized stool banks and various routes of administration, including capsules, enemas, or colonoscopies, offers opportunities to harness FMT as a potentially convenient and effective therapy for the treatment of NDDs. However, FMT treatment poses significant unique, and complex challenges for clinicians and regulators, including poorly defined mechanisms of action, stool availability, donor selection, adverse effects, and a relative lack of long-term follow-up data [164]. A recent study has summarized the limitations of previous FMT research, offering valuable insights to facilitate the actual clinical application of FMT. For example, special attention should be devoted to the distinctions between the human patient’s gut, harboring a complex gut microbiota, and the GF mouse gut, devoid of a resident microbiome; the relationship between the degree of transplanted colonization and therapeutic efficacy is a pivotal question that necessitates consideration; a holistic approach to the assessment of treatment outcomes, as opposed to a focus on isolated factors, is imperative [165]. The standardization of technical procedures, safety assessment, stool bank services, and management, among other aspects, is still in its infancy and necessitates further investigation.

## 5. Perspectives

Continuous advancements in the field of microbiome research are bringing us closer to unraveling the intricate connections between gut microbiota and their symbiotic relationship with humans. In the future, a comprehensive understanding of the impact of the microbiome on NDDs such as AD and PD will significantly contribute to their prevention, treatment, and management.

It is intriguing that apparently, simple daily dietary interventions may hold substantial therapeutic potential. The development of functional foods containing prebiotics and probiotics seems to have profound implications. Furthermore, as our understanding of the gut microbiota deepens, innovative strategies for noninvasive prognostic indicators and predictive biomarkers for various NDDs and their outcomes can be developed. For instance, using fecal DNA sequencing to quantify the richness of microbial communities and assess the ratio of “beneficial” to “harmful” bacteria may offer valuable insights. Ultimately, the uniqueness of individual microbiomes and targeted improvements in individual gut microbiota health will drive advances in personalized medicine.

## Figures and Tables

**Figure 1 nutrients-15-04631-f001:**
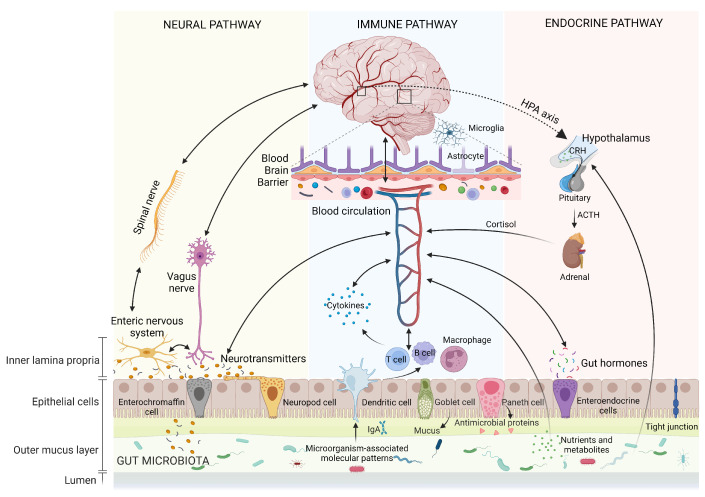
Pathways involved in bidirectional communication within the gut–brain axis (GBA). They include neural, immune, and endocrine pathways. Neurotransmitters: dopamine, serotonin, norepinephrine, gamma-aminobutyric acid (GABA), etc. Cytokines: interleukin (IL)-1β, IL-6, IL-10, tumor necrosis factor-α (TNF-α), etc. Nutrients and metabolites: short-chain fatty acids (SCFAs), amine compounds, vitamins, neuroprecursors, etc. ACTH: adrenocorticotropic hormone; HPA: hypothalamic-pituitary-adrenal; CRH: corticotrophin-releasing hormone. Created with BioRender.com.

**Figure 2 nutrients-15-04631-f002:**
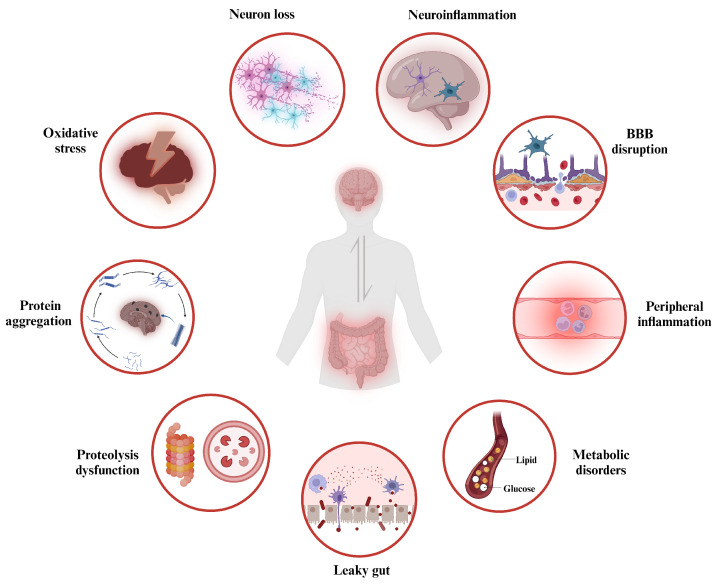
Hallmarks in the pathophysiology of neurodegenerative disorders caused by gut dysbiosis. Gut dysbiosis tends to promote the generation of proinflammatory cytokines and toxic metabolites, which in turn disrupt the integrity of the intestinal barrier, commonly referred to as “leaky gut”, and leads to an increased systemic circulation of inflammatory factors, microbes, and microbial products, thereby inciting systemic inflammation. Furthermore, the inflammatory status disrupts the blood–brain barrier (BBB), facilitating the entry of toxic metabolites like lipopolysaccharides (LPS) and β-N-methylamino-L-alanine (BMAA) into the brain, resulting in neuroinflammation and oxidative stress. Chronic neuroinflammation fosters the aggregation of pathological proteins, disrupting neuronal function and ultimately causing neuronal loss. Additionally, gut dysbiosis-induced chronic inflammation and oxidative stress impair autophagic clearance processes in both the gut and the brain, resulting in proteolysis dysfunction. Created with BioRender.com.

**Figure 3 nutrients-15-04631-f003:**
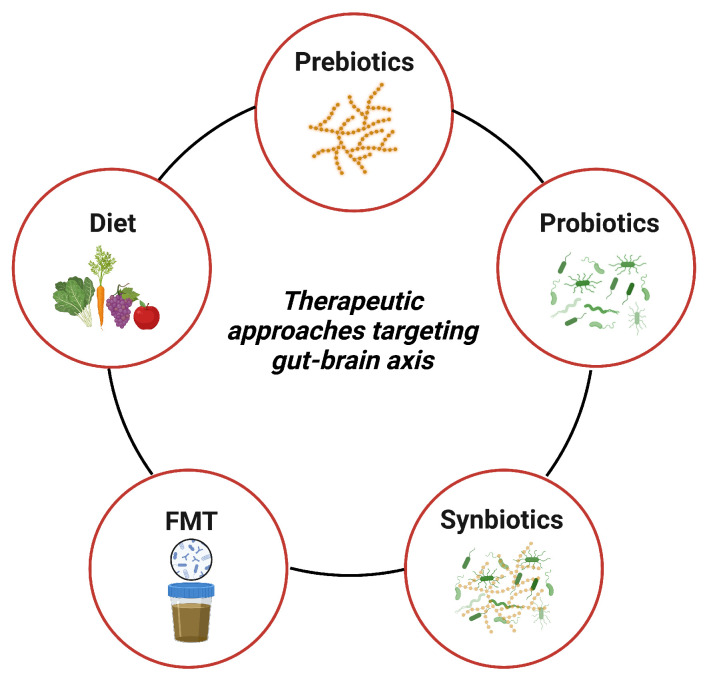
Strategies to modify gut microbiota for neurodegenerative disorders treatment. They mainly include diet, prebiotics, probiotics, synbiotics, and fecal microbiota transplantation (FMT). These approaches primarily function by modifying microbial communities and producing microbial metabolites, such as neurotransmitters and SCFAs, to exert neuroprotective effects. The diet emphasizes the consumption of fruits, vegetables, legumes, and cereals. Prebiotics are compounds selectively utilized by beneficial gut microbes, promoting the growth of beneficial bacteria and the generation of their metabolic products. Probiotics are live, nonpathogenic microorganisms that confer health benefits when consumed in adequate amounts. Synbiotics are specialized formulations that combine prebiotics and probiotics, synergistically enhancing their viability and therapeutic effects. FMT aims to restore a healthy gut microbiome and enhance gut microbiota diversity and functionality by transferring rigorously screened donor fecal microbiota into the patient’s GI tract. Created with BioRender.com.

## Data Availability

Not applicable.

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
