# Peer review of "Understanding the Gut–Brain Axis and Its Therapeutic Implications for Neurodegenerative Disorders"

_nutrients, 2023, doi:10.3390/nu15214631_

Round 1

Reviewer 1 Report

Comments and Suggestions for Authors

The present work reviews the relation of gut microbiota and neurodegenerative disorders in the context of Gut-Brain axis. In addition, the author showed several therapeutics interventions that modulate the gut microbiota. These interventions has showed a promised treatment in NDDs.

The data presented in this review is outstanding, however, minor corrections must be performed.

1. It is necessary to expand the legend of the figure 2. I suggest the addition of a small paragraph indicating the relation between the pathophysiological hallmarks and microbiota disbiosis. The idea is to understand the figure without an extensive analysis of the text of the review.

2. It is necessary to expand the legend of the figure 3. I suggest the addition of a small paragraph indicating briefly the therapeutic approaches indicated in the figure. The reason of my suggestion is based in the idea previously presented in the correction of figure 2.

3. As conclusion, the author comments that the dietary intervention may have therapeutic potential in many field such as neurodegenerative diseases. This point is crucial because maybe the dietary preferences in different populations could be related to NDDs indicators. Is there a relation between a healthy diet and less prevalence of NDDs?  For example, people following a mediterranean diet could be a low probability to develop a NDD compared with people following a high-fat diet?. I suggest the addition of a small paragraph comment this relevant point in the text. I think this discussion may improve enormously this review.

Reviewer 2 Report

Comments and Suggestions for Authors

The author has written an organized, thorough review of the research evidence describing a gut-brain axis underlying neurodegenerative diseases. A large number of mostly current references from reputable journals are cited.

Several corrections to sentence structure and verb tense are needed.

Suggestions for revisions:

Line 14: "I" rather than "we" provide.

95: are transmitted

122,123: intestinal hormones, or neurotransmitters...mechanical, chemical, or hormonal signals - use "and" instead of "or."

134: cognitive functions

166: TLR-deficient

175-177 and 188-189 and 269-270 and 564: sentences need correction

195: both excessive and

562: AB1-42-induced...

562: contributed

588: meaning of metabolic "status" is not clear

607: administration

616: a Drosophila model

4.5. Fecal microbiota transplantation line 622-

consider adding the concerns mentioned in: Phil. Trans. R. Soc. B 378: 20220221 https://doi.org/10.1098/rstb.2022.0221 such as questions about degree of colonization in fecal transplants.

Comments on the Quality of English Language

I list specific concerns in the comments to the author.

Round 2

Reviewer 2 Report

Comments and Suggestions for Authors

The author has revised the manuscript acceptably.